# Structural Changes, Electrophoretic Pattern, and Quality Attributes of Camel Meat Treated with Fresh Ginger Extract and Papain Powder

**DOI:** 10.3390/foods11131876

**Published:** 2022-06-24

**Authors:** Heba H.S. Abdel-Naeem, Mohammed M. Talaat, Kálmán Imre, Adriana Morar, Viorel Herman, Fathi A.M. El-Nawawi

**Affiliations:** 1Department of Food Hygiene and Control, Faculty of Veterinary Medicine, Cairo University, Giza 12211, Egypt; arafa2009@yahoo.com (M.M.T.); fathiel-nawawi@yahoo.com (F.A.M.E.-N.); 2Department of Animal Production and Veterinary Public Health, Faculty of Veterinary Medicine, Banat’s University of Agricultural Sciences and Veterinary Medicine “King Michael I of Romania”, 300645 Timişoara, Romania; adrianamo2001@yahoo.com; 3Department of Infectious Diseases and Preventive Medicine, Faculty of Veterinary Medicine, Banat’s University of Agricultural Sciences and Veterinary Medicine “King Michael I of Romania” from Timişoara, 300645 Timişoara, Romania; viorel.herman@fmvt.ro

**Keywords:** camel meat, ginger, papain, physicochemical, structural changes

## Abstract

Camel is a valuable source of meat for African and Asian countries; however, the most important problem associated with camel meat is its extreme toughness. This toughness has been attributed to its contents of connective tissue, which become more crossly linked in old animals. The toughness of camel meat decreases the consumer acceptance of this meat and reduces its chances of being utilized as a raw material for further processing into meat products. Ginger and papain were used in the current study as tenderizing enzymes, and the structural changes, electrophoretic pattern, physicochemical characteristics, and sensory scores of the treated meat were examined. The treatment of camel meat with ginger and papain resulted in marked destructive changes in the connective tissue and myofibers, and a reduction in the protein bands, with a consequent improvement in its tenderness. All the enzyme-treated samples exhibited significant increases in the protein solubility, with significant decreases in the shear-force values. Moreover, an improvement in the sensory scores of the raw and cooked meat and a reduction in the bacterial counts after the treatments were recorded. Ginger and papain induced a significant improvement in the physicochemical characteristics, sensory attributes, and bacterial counts of the camel meat; therefore, these materials can be utilized by meat processors to boost the consumer acceptance of this meat, and to increase its suitability as a raw material for further meat processing.

## 1. Introduction

Camel is an important source of meat production in Asia and African countries, and it has been considered an efficient animal for the production of meat, in comparison with other meat-producing animals. There are more than 25 million camels in the world, and the potential of camel products worldwide is continuously developing [1]. An extensive amount of meat from camel carcasses can be obtained for human consumption. Camel meat has many characteristics, which make this meat a superior and healthy food for humans, when compared with other meat sources. These characteristics include lower fat and cholesterol contents, with high polyunsaturated fatty acids, high essential amino acid-rich protein, a high moisture content, as well as high vitamin levels, and especially the vitamin-B complex [2]. Moreover, camel meat has the lowest total bacteria counts when compared with mutton, beef, and chicken, and it is free from pathogenic bacteria [3].

Although camel meat is valuable and superior to other sources of meat, the most critical issue connected to such meat is its excessive toughness at the retail level. The toughness of camel meat has been attributed to its contents of connective tissue. Camel meat is frequently derived from old animals that have performed different functions during their lives, and their connective-tissue fibers have become more crossly linked, with a consequent increase in the toughness of this meat [4]. The toughness of camel meat decreases the consumer acceptance of such meat and reduces its chances of being used as a raw material for further processing into meat products. Therefore, different methods have been adopted for tenderizing camel meat to improve its consumer acceptance and increase its suitability for further processing into different meat products.

The meat-tenderization process is mainly enzymatic, and it involves the utilization of microbial or plant enzymes [5]. The tenderization of meat with proteolytic enzymes is becoming a popular method, and, recently, proteolytic enzymes derived from the ginger rhizome (*Zingiber officinale* Roscoe) were introduced as effective enzymes for tenderizing the tough meat that arises from old animals [6].

The tenderizing effect of plant proteolytic enzymes has been verified [6,7,8]; however, the records of the effect of these enzymes on tougher meat, such as camel meat, are still limited. Moreover, the majority of previous meat-tenderization studies focused on the effect of such enzymes on the physicochemical characteristics of meat; meanwhile, examinations of the structural changes and the electrophoretic pattern of the meat after treatment with these enzymes are few. Accordingly, the goal of the current study was to reveal the structural changes (using a light microscope, transmission electron microscope, and scanning electron microscope) and electrophoretic pattern of fresh camel meat after treatment with ginger and papain. Moreover, maintaining the quality attributes of meat is a critical issue for meat processors; therefore, the physicochemical properties, sensory attributes, and bacterial counts of the treated meat were assessed. This work may increase the consumer acceptability of camel meat and promote meat processors to utilize camel meat as raw-meat material during the production of good-quality camel-meat products.

## 2. Materials and Methods

### 2.1. Experimental Design

A three-replicate-based experiment (three replicates at different times, with three samples at each replicate) was conducted to examine the effect of ginger extract (7%), papain powder (0.7%), as well as a mixture of ginger extract (5%) and papain powder (0.5%) on the structure, electrophoretic pattern, and quality attributes of the *semitendinosus* muscle of 8-year-old camel (“*Camelus dromedarius* L. (1758)”) meat.

### 2.2. Enzyme Preparation

Fresh ginger rhizome (*Zingiber officinalis Roscoe*) was obtained from a local market and was prepared according to the method described by Abdel-Naeem and Mohamed [1]. Papain-enzyme powder (Loba Chemie Mumbai, India) was utilized and, just prior to its application, the required concentration was prepared by dissolving it in distilled water.

### 2.3. Muscle-Sample Preparation

*Semitendinosus* muscles of both sides of ~8-year-old female camel (“*Camelus dromedarius* L. (1758)”) carcasses were obtained 4 h after slaughter. Animals were slaughtered at the El-Bassateen municipality slaughterhouse of the Cairo Governorate, Cairo, Egypt, after having been held in a lairage for 12 h, and were dressed following a routine traditional halal procedure. *Semitendinosus* muscle samples were placed in polyethylene bags and stored for 20 h at 4 °C. After an elapse of about 24 h postslaughter, muscles were trimmed of visible fat, connective tissue was then cut into uniform-sized chunks (3 × 3 × 3 cm), and they were exposed to different treatments.

### 2.4. Enzyme Treatment and Marination

Approximately 27 cm^3^ uniform-sized camel-meat chunks were mixed with fresh ginger extract (7% *w*/*v*), papain-enzyme powder (0.7% *w*/*w*), and a mixture of fresh ginger extract (5% *w*/*v*) and papain-enzyme powder (0.5% *w*/*w*). A control treatment was prepared by mixing meat chunks with distilled water. Meat chunks were packed in polyethylene bags after thorough hand mixing, and were then stored at 4 °C for 48 h. Thereafter, four treatments were prepared as follows: (a) control: meat chunks mixed with 15 mL distilled water; (b) ginger (7% *w*/*v*): meat chunks mixed with 7 mL fresh ginger extract + 8 mL distilled water; (c) papain (0.7% *w*/*w*): meat chunks mixed with 0.7 g papain-enzyme powder + 14.3 mL distilled water; (d) ginger (5% *w*/*v*) + papain (0.5% *w*/*w*): meat chunks mixed with 5 mL fresh ginger extract + 0.5 g papain-enzyme powder + 9.5 mL distilled water. Finally, meat chunks were drained after marination for 48 h and were then exposed to structural, physicochemical, bacterial, and sensory analyses.

### 2.5. Investigations

#### 2.5.1. Quality Attributes

##### Measurement of Collagen Solubility, Sarcoplasmic Protein Solubility, and Myofibrillar Protein Solubility

Collagen solubility was examined according to the method recommended by Naewbanij et al. [9]. However, the solubility of the sarcoplasmic and myofibrillar protein was measured using the method of Joo et al. [10].

##### Measurement of Myofibrillar Fragmentation Index

Cubes of 7 mm were prepared from the samples after the removal of fat and epimysia connective tissue, and the myofibrillar fragmentation index (MFI) was calculated [11].

##### Measurement of Sarcomere Length and Muscle-Fiber Diameter

Muscle-tissue core (2.54 cm) was fixed for 24 h in formal saline, and was blended for 30 s at low speed, using a lab blender. A drop of the homogenate was placed over a glass slide, covered with a coverslip, and examined under a light microscope (Labomed, Inc., Los Angeles, CA, USA) with a 10× eyepiece containing a calibrated micrometer. Sarcomeres lengths (SLs) and muscle-fiber diameters (MFDs) of 21 muscle fibers were measured, and the average of each measurement was expressed in microns [12].

##### The pH Value

Five grams from each sample were homogenized with 20 mL distilled water for 10–15 s. The pH was measured using a pH meter (SensoDirect 150, Lovibond, Sarasota, FL, USA) with a probe-type electrode (Lovibond Senso Direct Type 330, FL, USA). Three readings were taken from each sample, and then the average was recorded. The meter was adjusted using two buffers (7.0 and 4.0) between every two samples [13].

##### Thiobarbituric Acid Reactive Substances

Thiobarbituric acid reactive substances (TBARS) were measured according to the method of Du and Ahn [14], and they were expressed as milligrams of malonaldehyde per kilogram of the sample.

##### Cooking Loss

Cooked meat samples were cooled after cooking in the water bath at 100 °C for 35 min to achieve a 75 °C core temperature, and they were reweighted to determine the cooking-loss percentage from the weight, before and after cooking.

##### Shear Force

From water-bath-cooked meat blocks, six core samples, with diameters of 1.27 cm, were removed by a handheld core device, parallel to the longitudinal axis of muscle fibers. All samples were sheared with a Warner–Bratzler shear-force (WBSF) machine connected to an Instron Universal Testing device (Model 2519-105, Instron Corp., Canton, MA, USA), with a 200 mm/min crosshead speed, and a tension/compression load of 55 kg. For each sample, the average shear-force value was obtained and calculated [15].

##### Color Evaluation

Meat surface color was examined using a Chroma meter (Konica Minolta, model CR 410, Marunouchi, Japan), which was calibrated against a white calibration plate. The L* (lightness), a* (redness), and b* (yellowness) values were obtained using the CIE standard illuminant D_65_ light source, with a bloom time of 30 min and a 10° observation angle. From each sample surface, three readings were obtained, and the average was recorded and expressed according to the Commission International de l’Eclairage (CIE) (L*, a*, b*) color system [16].

##### Bacterial Counts

For bacterial counts, tenfold serial dilution was prepared, and standard Plate Count Agar (PCA) (Oxoid CM0463, Basingstoke, UK) was inoculated. The inoculated plates were incubated at 35 °C for 48 h for the enumeration of aerobic plate counts [17], and at 7 °C for 7 days for the enumeration of psychrotrophic bacteria [18].

##### Sensory Panel Analysis

Sensory evaluation for all samples was examined, following the guidelines of the AMSA [19]. Nine experienced panelists were chosen from the staff members of the Department of Food Hygiene and Control at Cairo University, Egypt. Panelists received a preparatory session before testing. All testing was carried out under controlled conditions. Tap water was provided between samples, to cleanse the palate. Sensory analysis was conducted on raw and cooked meat chunks. For raw samples, the panelists were asked to score each sample for appearance, odor, consistency, and overall acceptability, using 9-point numerical scores, with 9 being highly desirable, and 1 being highly undesirable. For cooked samples, chunks were placed in polypropylene bags, sealed, and cooked by moist heat using a water bath (Kottermann D-3165, Uetze, Germany) at 100 °C for 35 min to achieve a 75 °C core temperature, monitored by a handheld thermometer (Hanna HI 985091-1, Wensocket, RI, USA). In a randomized order, the panelist assessed three replicates from all treatments and was asked to give a value from 1 (extremely unacceptable) to 9 (extremely acceptable) for the following sensory scores: appearance, flavor, juiciness, tenderness, and overall acceptability.

#### 2.5.2. Structural Examination

##### Light Microscope

Camel *semitendinosus* muscle samples (1 × 1 cm) were fixed for 24 h in formalin (10%) and were then prepared and stained with Hematoxylin and Eosin [20]. From each treatment, at least 40 micrographs were examined to reveal the changes that occurred following treatments.

##### Transmission Electron Microscope

Muscle samples were cut into 1 × 1 × 3 mm pieces, and they were fixed for 2 h at 4 °C by immersion in 2.5% glutaraldehyde in 0.1 M phosphate buffer (pH 7.2), while taking into account the muscle-fiber orientation. Fixed samples were prepared for transmission electron microscope [21]. Ultrathin sections of 50 nm were cut using an ultramicrotome (SEO UMTP-6, Moscow, Russia), stained with aqueous uranyl acetate and alkaline lead citrate, and examined with the transmission electron microscope (SEO TEM-100, Moscow, Russia), operated at an accelerating voltage of 80 kV.

##### Scanning Electron Microscope

Muscle samples were cut into small pieces (2 × 2 × 3 mm) and immediately fixed in phosphate-buffered glutaraldehyde (2.5%) for 2 h at 4 °C. Samples were rinsed three times with phosphate buffer saline (0.1 M) for 30 min per time, and were then dehydrated in 50–100% of ethanol (15 min for each concentration). Samples were then dried using a critical point drier (Samdri PVT-3D, Tousimis, Rockville, MD, USA) for 15 min, followed by gold coating in a vacuum evaporator (JFC 1100 E, Jeol Ltd., Tokyo, Japan), and were examined under scanning electron microscopy (JSM 5300, Jeol Ltd., Tokyo, Japan) [22].

#### 2.5.3. Electrophoresis

Sodium dodecyl sulphate–polyacrylamide gel electrophoresis (SDS–PAGE) was carried out with an Omni PAGE Maxi, Stock (VS20-48-1) electrophoresis apparatus [23]. Five grams of minced meat samples were mixed with 50 mL of 0.01 N sodium phosphate buffer (pH 7.0), containing SDS (1%) and 2-mercaptoethanol (1%), and were then incubated at 37 °C for 2 hours. The mixture was centrifuged at 1500 g for 30 min. An aliquot of the supernatant was dialyzed overnight at room temperature against 0.1 N sodium phosphate buffer containing 2-mercaptoethanol (0.1%). About 50 µm of the dialyzed solution was used for loading the gel. Electrophoresis was performed at a constant voltage mode of 100 V/slab, at 30 mA, until the tracking dye reached the lower end of the gel (5–6 h). The gel was removed and stained with Coomassie blue for 4–5 h, and then the gel was destained and photographed.

### 2.6. Statistical Analysis

Statistical analysis for all data was conducted using SPSS statistics for Windows, version 17.0 (IBM Corp., Armonk, NY, USA). One-way analysis of variance (ANOVA) was used to compare the data, and the least-squares-difference (LSD) test was used to determine the significance. *p* < 0.05 was considered significant. Correlation analyses for the relations between some sensory attributes (tenderness and juiciness) and other quality parameters (collagen content, collagen solubility %, total soluble protein, myofibrillar fragmentation index, muscle-fiber diameter, sarcomere length, and shear force) of all enzyme-treated samples were performed.

## 3. Results and Discussions

### 3.1. Quality Attributes

The results of the quality attributes of the camel-meat samples treated with the fresh ginger extract (7%), papain enzyme (6%), and a combination of fresh ginger extract (5%) and papain powder (0.5%) are presented in Table 1. The collagen, sarcoplasmic protein, and myofibrillar protein solubility values of all the enzyme-treated samples were significantly (*p* < 0.05) higher than the counterpart control samples. It is noteworthy that, among all the enzyme-treated samples, the samples treated with the mixture of ginger extract and papain enzyme had the highest values, followed by the papain-treated samples, while the lowest values were recorded for the samples treated with the ginger extract (Table 1). The obtained findings are in agreement with previous studies on different types of meat [7,24,25,26,27].

The increase in the protein solubility of all the enzyme-treated meat samples in this study may be attributed to an increase in the connective-tissue and myofibrillar protein permeability, and, as a consequence, they disintegrate easily. Likewise, proteases act on the intermolecular crosslinks of collagen fibers, which causes structural alterations [28].

The obtained findings revealed that the myofibrillar fragmentation index and sarcomere-length values were significantly (*p* < 0.05) increased nonetheless, and the muscle-fiber-diameter values were significantly (*p* < 0.05) decreased in all the enzyme-treated samples (Table 1). Furthermore, all the enzyme-treated samples exerted a significant (*p* < 0.05) increase in the cooking loss %, and significant (*p* < 0.05) decreases in the pH, TBARS, shear-force, and a* values, with nonsignificant (*p* ˃ 0.05) changes in the values of L* and b*, in comparison with the untreated control samples (Table 1).

On the one hand, the data on the cooking loss were in harmony with those of Abdeldaiem and Ali [26], who recorded a significant reduction in the cooking loss in camel meat treated with ginger. On the other hand, Naveena et al. [7] recorded a nonsignificant reduction in the cooking loss of ginger- and papain-treated buffalo meat. The significant increase in the cooking loss in all the enzyme-treated samples, in the current study, is due to the proteolytic effect of such enzymes, which induces protein denaturation [6,7,29]. In addition, the change in the pH values of the camel meat treated with ginger, papain, and a combination of them was explained by their effect on the ionic strength of the treated meat. Insignificant increases in the pH values were recorded in ginger-treated goat meat [25] and ginger- and papain-treated buffalo meat [7]. Higher pH values were noticed in ginger-extract-treated buffalo meat [30]. However, a nonsignificant reduction in the pH values was recorded in camel meat treated with ginger extract [26]. Our results regarding the reduction in the TBARS of the ginger-treated samples, in the current study, are consistent with those reported by Mendiratta et al. [24] for sheep meat, Pawar et al. [25] for goat meat, and Abdel-Naeem and Mohamed [1] for camel burger. The antioxidant effect of ginger was elucidated by its peroxide-scavenging activity, which reduces the oxidation of unsaturated fatty acids. Moreover, ginger contains some components that have desaturase and elongase activities [31].

Significant reductions in the shear-force values were noticed in ginger-extract-treated meat and meat products [1,25,26], and in papain-treated meat [7]. The tenderizing activity of these enzymes, in addition to their ability to solubilize collagen, may be the cause of the reduction in the shear-force values of the treated samples. The reduction in the a* values of the treated samples may be due to the effect of these tenderizing enzymes on the pigments of the treated meat. Singh et al. [32] obtained nonsignificant changes in the L*, a*, and b* values of a chicken-meat emulsion treated with ginger. A significant increase in the L* values of ginger-treated camel-meat burger, and nonsignificant changes in the color values (L*, a*, and b*) of papain-treated camel-meat burger, were previously obtained [1].

#### 3.1.1. Bacterial Counts

The treatment of camel meat with fresh ginger extract, papain enzyme, and their combination resulted in a significant (*p* < 0.05) reduction in the aerobic plate count (APC) and psychrotrophic bacterial counts (Table 1). A reduction in the total bacterial counts of pork burgers treated with ginger powder, at concentrations of 1 and 2% during storage, has been reported [33]. The antibacterial effect of ginger and its extracts has been recorded against different microorganisms, including *Escherichia coli*, *Salmonella typhi*, *Enterobacter* spp., *Klebsiella* spp., and *Pseudomonas aeruginosa* [34,35,36]. Moreover, it has been reported that ginger root contains several compounds that exert antimicrobial properties [37,38].

#### 3.1.2. Sensory Evaluation

The sensory scores of the raw and cooked camel meat from the treated samples and control are shown in Table 2.

The appearance, odor, consistency, and overall acceptability scores of the raw camel meat treated with ginger (7%), papain (0.7%), and a combination of ginger (5%) and papain (0.5%) were significantly (*p* < 0.05) higher than those of the control samples. The odor scores of the raw camel meat treated with ginger alone, or with a combination of ginger and papain, were significantly (*p* < 0.05) higher than the papain-treated samples. The appearance, flavor, juiciness, tenderness, and overall acceptability scores of the cooked camel meat treated with ginger, papain, and combination of them were significantly (*p* < 0.05) higher than those of the control. The appearance and juiciness values of the papain-treated samples were significantly (*p* < 0.05) lower than those of the samples treated with ginger only, or those treated with a combination of ginger and papain; however, the tenderness values of the papain-treated samples were significantly (*p* < 0.05) higher than those of the ginger-treated samples. The tenderness scores of all the treated samples were in accordance with the findings of the shear-force values. Improvements in the sensory scores of different ginger- and/or papain-treated meat and meat products have been recorded [1,6,26,39].

The correlation matrix between the tenderness or juiciness with some of the quality parameters of the camel *semitendinosus* meat samples is presented in Table 3.

Both the tenderness and juiciness induced a strong correlation (*p* < 0.05) with the collagen content, collagen solubility %, total soluble protein, MFI, MFD, SL, and shear force in all the enzyme-treated samples. Therefore, the proteolytic influence of the different enzymatic treatments with improving tenderness is owed to its destructive effect on both connective-tissue and muscle-fiber elements, along with its great action on the solubility of different types of proteins, and especially collagen.

### 3.2. Structural Changes

#### 3.2.1. Light Microscope

Light micrographs of the control camel semitendinosus muscle stained with H&E revealed straight, closely bound, intact, nucleated muscle fibers, with a huge amount of intact fibrous connective tissue (Figure 1: C).

The treatment of camel meat with fresh ginger extract (7%) resulted in the appearance of multiple cracks along the muscle fibers and increased spaces between the muscle fibers, in addition to the slight destructive change in the connective tissue (Figure 1: G). Nonetheless, light micrographs of the camel meat treated with papain enzyme (0.7%) revealed a more destructive effect in the muscle fibers, with an obvious change in the fibrous connective tissue (Figure 1: P). However, the micrographs of muscle samples treated with a mixture of fresh ginger extract (5%) and papain (0.5%) displayed severe muscle fragmentation, along with the extensive degradation of connective tissue (Figure 1: Mix). Such findings are in good agreement with those observed in camel-meat burgers [1].

The none-stained camel semitendinosus muscle samples revealed that all the enzymatic treatments of the camel-meat samples showed severe muscle-fiber fragmentation, in comparison with the control samples, which remained intact (Figure 2: G, P, and Mix). Furthermore, it is important to highlight that all the enzyme-treated samples induced marked changes in both the MFD and SL, which were manifested by a reduction in the diameter of the muscle fiber and an increase in the sarcomere length, which improved their tenderness scores (Table 1).

#### 3.2.2. Transmission Electron Microscope and Scanning Electron Microscope

Transmission electron micrographs of the control camel semitendinosus muscle sample at magnifications of ×20,000, ×40,000, and ×80,000 revealed the normal alignment of the Z-line, M-line, I-band, and A-band, with the normal narrowing of these two bands, in addition to the bulging of the sarcomeres from the centers. Moreover, the Z-line and the junctions between the I-band and Z-line were intact, without any changes (Figure 3: C). Nevertheless, the treatments of camel meat with ginger (Figure 3: G), papain (Figure 4: P), and their combination (Figure 4: Mix) resulted in pronounced breaks, fragmentations across the myofibrils, the degradation and dissolving of the Z-line materials, the degradation of the junctions (J) between the Z-line and I-band, as well as multiple cleavages in both the A- and I-bands. Such effects were more pronounced in the samples treated with the mixture of both ginger and papain extracts (Figure 4: Mix), followed by those treated with papain (Figure 4: P); however, minimal changes were noticed in the ginger-extract-treated samples (Figure 3: G).

It is noteworthy to mention that studies that illustrate the effect of ginger and papain enzymes on the microstructure of meat samples as tenderizing agents are very limited. To the best of our knowledge, this study is the first regarding the effect of ginger and papain enzymes on the ultrastructures of camel meat. However, the preferential fragmentation of myofibrils, with the severe degradation of the thin filaments in the I-bands, was revealed previously in transmission electron micrographs of ginger-treated beef meat [40]. Furthermore, slightly deformed Z-discs with stretched sarcomeres were observed in the transmission electron micrographs of ammonium hydroxide-treated buffalo samples [41].

Scanning electron micrographs of control camel *semitendinosus* muscle revealed the presence of a dense connective-tissue sheet that completely covered the structure beneath, and, in some areas, the muscle fibers appeared closely packed to each other, with narrow intermyofiber spaces in between (Figure 5: C). The treatment of camel meat with ginger (Figure 5: G), papain (Figure 6: P), and a mixture of them (Figure 6: Mix) resulted in significant changes in the connective-tissue structure, and the appearance of cracks across the lengths of the muscle fibers.

These changes were more prominent in the samples treated with the mixture of both ginger extract and papain enzyme (Figure 6: Mix), in which the connective tissue was replaced by an undefined amorphous mass, and several cracks and cleavages appeared along the lengths of the muscle fibers, with a wavy appearance.

The obtained findings are in agreement with those reported by Naqvi et al. [42], who found that the treatment of beef-muscle samples with ginger powder induced the tearing and severe degradation of the perimysium of raw samples, which is difficult to distinguish after cooking. Furthermore, the scanning electron micrographs of ginger-treated buffalo meat revealed fragmentation in the myofibrillar structure, with high exudates, and the development of gaps among the muscle fibers [30].

The authors explained such findings by the breakdown of the endomysial collagen and connective tissue around the muscle fibers. The topographical structure of spent hen-breast-meat samples marinated with ginger extract revealed endomysium degradation, with a subsequent improvement in their tenderness [43]. Likewise, the extensive degradation of endomysial connective-tissue layers surrounding the myofibers, with wide gaps between them, was noticed in papain-enzyme-treated beef samples [28]. In another study, Zhao et al. [44] observed needles, or short bars, that cut the muscle fibers in papain-treated beef samples.

### 3.3. The Electrophoretic Pattern of Muscle Proteins

A representative SDS-PAGE gel for the control and enzyme-treated camel-meat samples is presented in Figure 7.

The electrophoretic pattern of all the enzyme-treated muscle proteins revealed a reduction in the number of protein bands, which indicates the increased proteolysis of muscle proteins (Figure 7: G, P, and Mix). Moreover, the electrophoretic pattern revealed the breakdown or cleavage of the higher-molecular-weight proteins into lower-molecular-weight proteins. The protein degeneration was more obvious in the samples treated with a mixture of ginger and papain, which indicated more protein proteolysis (Figure 7: Mix). The protein proteolysis was strongly linked to higher protein solubility in the enzyme-treated meat samples (Table 1). Our results are in agreement with those reported by Abdeldaiem and Ali [26] for ginger-extract-treated camel meat. Furthermore, the pattern of the breakdown of high-molecular-weight proteins into low-molecular-weight proteins is in consistent with the observations of Ha et al. [29]. In the same regard, Zhao et al. [44] found that the treatment of beef with papain at 37 °C for 1 h degraded all the myofibrillar proteins with molecular weights lower than 20 kDa.

## 4. Conclusions

From this study, it can be concluded that the treatment of camel meat with fresh ginger extract and papain enzyme produced the degradation of muscle fibers and the destruction of connective tissue, which resulted in a tenderizing effect on this meat. These effects resulted in the improvement in the sensory attributes and physicochemical characteristics of the camel meat. These results may encourage meat processors to produce camel meat that can be accepted by consumers, and to utilize this meat as a raw material for meat processing.

## Figures and Tables

**Figure 1 foods-11-01876-f001:**
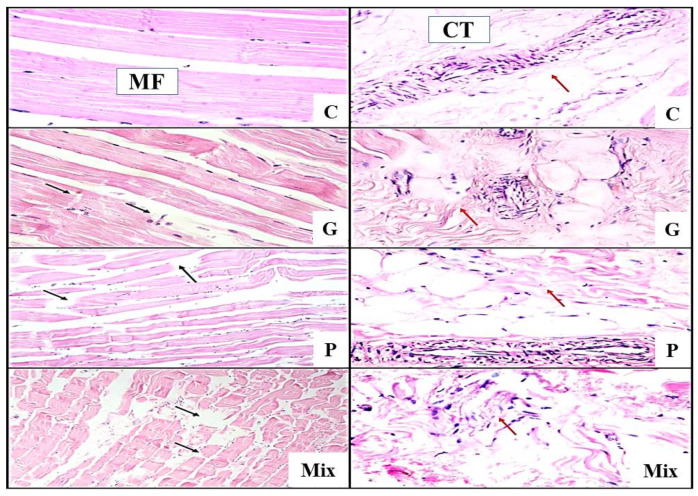
Light micrograph of camel *semitendinosus* muscle stained with H&E (×40). C: control; G: ginger extract; P: papain-enzyme powder; Mix: a mixture of ginger and papain; MF: muscle fiber; CT: connective tissue. Black arrows point to fiber breaks in muscle fibers; red arrows point to connective tissue.

**Figure 2 foods-11-01876-f002:**
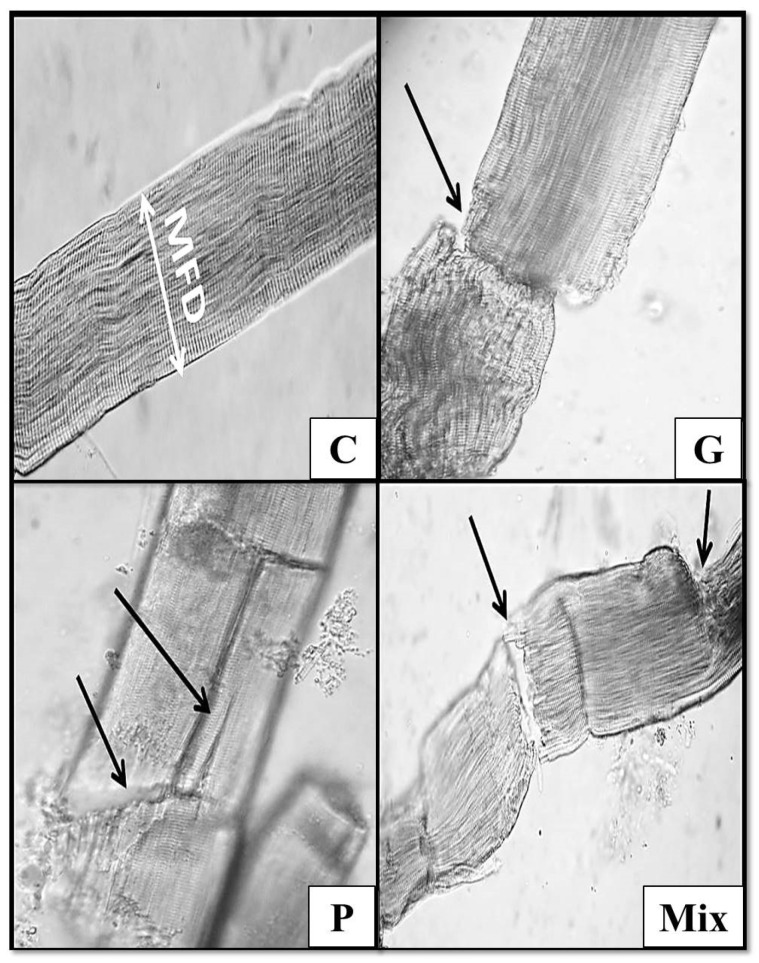
Light micrograph of none-stained camel semitendinosus muscle samples (×40). C: control; G: ginger extract; P: papain-enzyme powder; Mix: a mixture of ginger and papain. White arrow points to MFD (muscle-fiber diameter); black arrows point to fragmentation in muscle fibers.

**Figure 3 foods-11-01876-f003:**
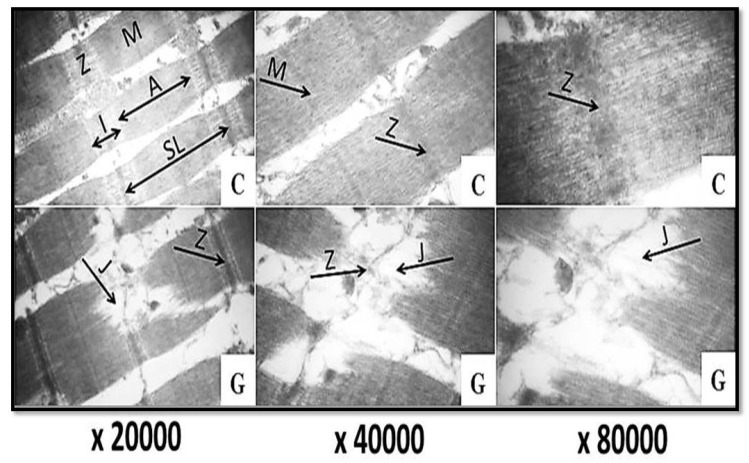
Transmission electron micrographs of camel semitendinosus muscle (×20,000, ×40,000, and ×80,000). C: control; G: ginger extract; A: A-band; I: I-band; SL: sarcomere length; Z: Z-line; M: M-line; J: junction between I-band and Z-line.

**Figure 4 foods-11-01876-f004:**
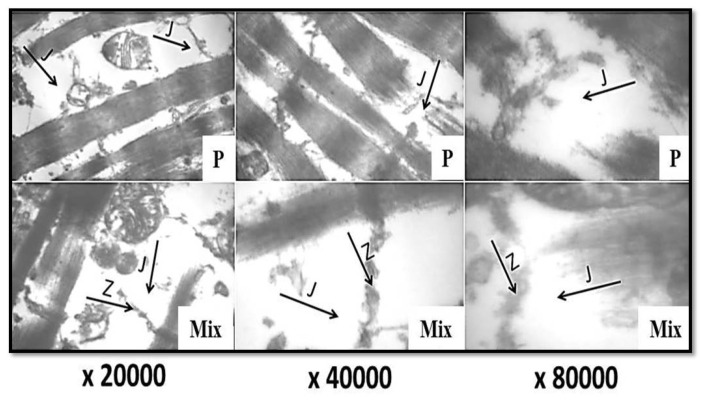
Transmission electron micrographs of camel semitendinosus muscle (×20,000, ×40,000, and ×80,000). P: papain-enzyme powder; Mix: a mixture of ginger and papain; Z: Z-line; J: junction between I-band and Z-line.

**Figure 5 foods-11-01876-f005:**
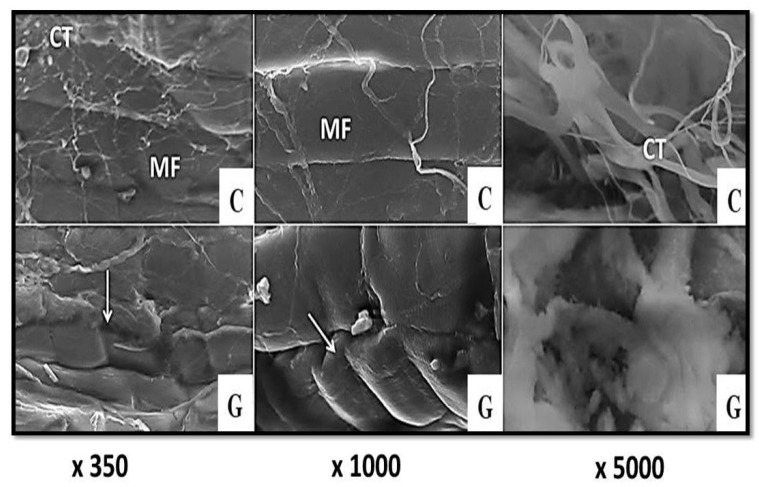
Scanning electron micrographs of camel semitendinosus muscle (×350, ×1000, and ×5000). C: control; G: ginger extract; MF: muscle fiber; CT: connective tissue. White arrows point to fragmented muscle fibers.

**Figure 6 foods-11-01876-f006:**
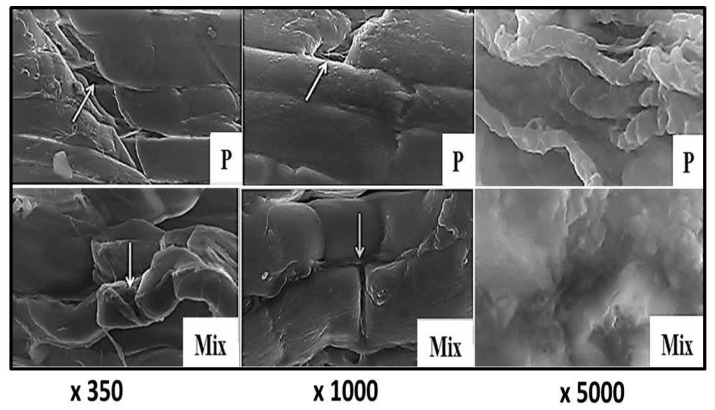
Scanning electron micrographs of camel semitendinosus muscle (×350, ×1000, and ×5000). P: papain-enzyme powder; M: a mixture of ginger and papain. White arrows point to fragmented muscle fibers.

**Figure 7 foods-11-01876-f007:**
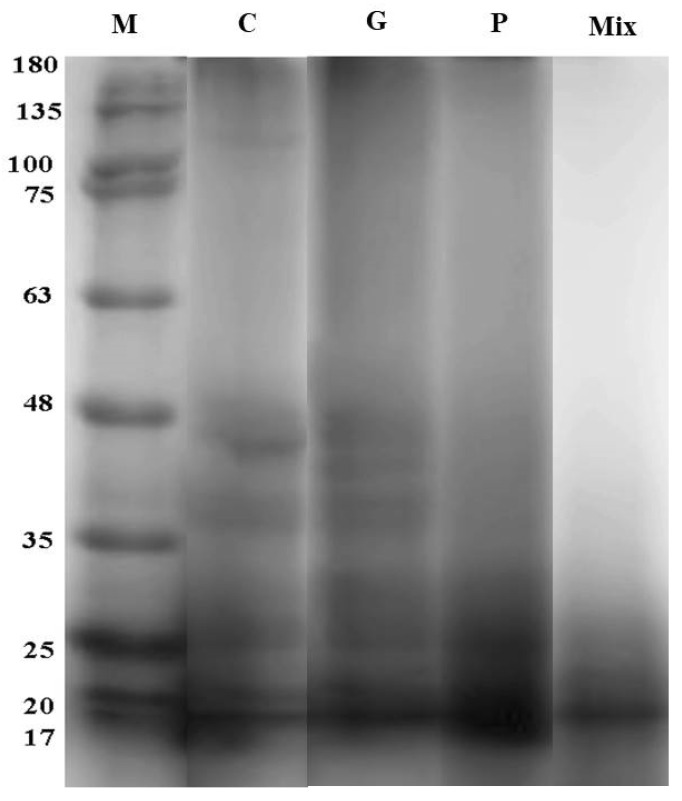
Electrophoretic pattern of camel *semitendinosus* muscle. M: protein marker; C: control; G: ginger extract; P: papain-enzyme powder; Mix: a mixture of ginger and papain.

**Table 1 foods-11-01876-t001:** Quality attributes of camel semitendinosus muscle after treatment with fresh ginger extract (7%), papain-enzyme powder (0.7%), and their combination (5% ginger and 0.5% papain).

Parameters	Treatments			
	Control	7% Ginger	0.7% Papain	5% Ginger + 0.5% Papain
Collagen solubility (%)	3.60 ^c^ ± 0.19 *	10.36 ^b^ ± 2.67	18.28 ^a^ ± 2.70	23.85 ^a^ ± 3.19
Sarcoplasmic protein solubility (mg/g)	1.85 ^b^ ± 0.17	2.73 ^a^ ± 0.18	2.91 ^a^ ± 0.21	3.45 ^a^ ± 0.14
Myofibrillar protein solubility (mg/g)	1.40 ^b^ ± 0.21	2.49 ^a^ ± 0.23	3.03 ^a^ ± 0.27	3.58 ^a^ ± 0.37
Myofibrillar fragmentation index%	69.00 ^b^ ± 0.87	86.00 ^a^ ± 1.76	78.33 ^a^ ± 0.37	82.44 ^a^ ± 1.48
Muscle-fiber diameter (µm)	113.40 ^a^ ± 3.15	59.40 ^b^ ± 5.85	77.40 ^b^ ± 4.95	61.65 ^b^ ± 6.30
Sarcomere length (µm)	2.61 ^b^ ± 0.09	4.50 ^a^ ± 0.18	3.38 ^a^ ± 0.14	4.10 ^a^ ± 0.18
pH	5.76 ^a^ ± 0.03	5.60 ^b^ ± 0.01	5.67 ^b^ ± 0.04	5.62 ^b^ ± 0.02
TBARS (mg/kg)	0.33 ^a^ ± 0.03	0.15 ^c^ ± 0.01	0.20 ^b^ ± 0.01	0.13 ^c^ ± 0.01
Cooking loss %	33.40 ^c^ ± 0.74	36.57 ^b^ ± 1.20	39.56 ^a^ ± 1.40	42.85 ^a^ ± 0.90
Shear force (N)	87.67 ^a^ ± 0.36	63.55 ^b^ ± 0.55	56.39 ^b^ ± 0.22	57.86 ^b^ ± 0.53
Color values				
L*	36.39 ^a^ ± 0.40	38.84 ^a^ ± 1.70	38.24 ^a^ ± 1.61	38.66 ^a^ ± 1.74
a*	18.87 ^a^ ± 0.26	14.08 ^b^ ± 0.81	15.04 ^b^ ± 1.26	15.41 ^b^ ± 1.07
b*	7.63 ^a^ ± 0.38	6.70 ^a^ ± 0.74	5.65 ^a^ ± 0.95	6.99 ^a^ ± 0.20
Bacterial counts				
Aerobic plate count (Log_10_ CFU/g)	6.23 ^a^ ± 0.06	5.08 ^b^ ± 0.05	5.39 ^b^ ± 0.04	3.97 ^c^ ± 0.13
Psychrotrophic (Log_10_ CFU/g)	5.70 ^a^ ± 0.03	5.00 ^b^ ± 0.03	5.02 ^b^ ± 0.01	4.40 ^c^ ± 0.05

^a–c^ Means with different superscripts within the same row for each parameter are significantly (*p* < 0.05) different. * Values represent the means of 3 independent replicates ±SE.

**Table 2 foods-11-01876-t002:** Sensory scores of camel semitendinosus muscle after treatment with ginger extract (7%), papain-enzyme powder (0.7%), and their combination (5% ginger and 0.5% papain).

Sensory Attributes	Treatments			
	Control	7% Ginger	0.7% Papain	5% Ginger + 0.5% Papain
Raw				
Appearance	4.67 ^b^ ± 0.24 *	6.89 ^a^ ± 0.20	6.33 ^a^ ± 0.17	7.00 ^a^ ± 0.24
Odor	4.00 ^c^ ± 0.17	7.50 ^a^ ± 0.10	6.22 ^b^ ± 0.15	7.33 ^a^ ± 0.29
Consistency	4.33 ^b^ ± 0.17	6.67 ^a^ ± 0.29	7.44 ^a^ ± 0.29	7.11 ^a^ ± 0.35
Overall acceptability	4.78 ^b^ ± 0.22	7.00 ^a^ ± 0.17	7.22 ^a^ ± 0.15	7.11 ^a^ ± 0.35
Cooked				
Appearance	5.85 ^b^ ± 0.14	7.07 ^a^ ± 0.16	5.96 ^b^ ± 0.23	6.63 ^a^ ± 0.13
Flavor	4.37 ^c^ ± 0.17	7.11 ^a^ ± 0.20	6.07 ^b^ ± 0.10	6.67 ^a^ ± 0.12
Juiciness	4.45 ^b^ ± 0.17	6.30 ^a^ ± 0.19	6.74 ^a^ ± 0.23	6.78 ^a^ ± 0.22
Tenderness	3.46 ^c^ ± 0.11	6.11 ^b^ ± 0.21	6.99 ^a^ ± 0.20	6.81 ^a,b^ ± 0.37
Overall acceptability	3.89 ^b^ ± 0.11	6.81 ^a^ ± 0.10	6.63 ^a^ ± 0.15	6.96 ^a^ ± 0.10

^a–c^ Means with different superscripts within the same row for each parameter are significantly (*p* < 0.05) different. * Values represent the means of 3 independent replicates ±SE.

**Table 3 foods-11-01876-t003:** Correlation matrix between some sensory attributes and other quality parameters of camel semitendinosus muscle tenderized with different enzymes.

Parameters	7% Ginger	0.7% Papain	5% Ginger + 0.5% Papain
Tenderness			
Collagen content	−0.844 **	−0.884 **	−0.887 **
Collagen solubility %	0.715 **	0.731 **	0.852 **
Total soluble protein	0.670 **	0.919 **	0.876 **
Myofibrillar fragmentation index	0.866 **	0.878 **	0.856 **
Muscle-fiber diameter	−0.772 **	−0.655 **	−0.761 **
Sarcomere length	0.860 **	0.722 **	0.769 **
Shear force	−0.647 **	−0.886 **	−0.733 **
Juiciness			
Collagen content	−0.883 **	−0.897 **	−0.903 **
Collagen solubility %	0.630 **	0.721 **	0.828 **
Total soluble protein	0.707 **	0.920 **	0.865 **
Myofibrillar fragmentation index	0.838 **	0.878 **	0.874 **
Muscle-fiber diameter	−0.806 **	−0.706 **	−0.792 **
Sarcomere length	0.867 **	0.740 **	0.806 **
Shear force	−0.676 **	−0.854 **	−0.702 **

The critical value at 5% = 0.444 ** The critical value at 1% = 0.561.

## Data Availability

Data is contained within the article.

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
