# Peer review of "Structural Changes, Electrophoretic Pattern, and Quality Attributes of Camel Meat Treated with Fresh Ginger Extract and Papain Powder"

_foods, 2022, doi:10.3390/foods11131876_

Round 1
Reviewer 1 Report
The manuscript entitled Structural changes, electrophoretic pattern, and quality attributes of camel meat treated with fresh ginger extract and papain powder has great scientific potential. Some major changes need to be made.
Page 2, line 68:reveal - incorrectly used word
Page 2, line 81: semitendinosus – italics
Page 2, line 90: semitendinosus – italics (correct throughout the manuscript)
Page 3, line 111: 2.5. Investigations 2.5.1. Physicochemical Criteria - incorrectly used word
Page 5, line 223: [7,24,25,26,27] - you don't need so many citations
Table 1: Collagen solubility (%) - what is the reason for such a big difference in collagen?
Control 3.60c ± 0.19
5% ginger+ 0.5% Papain 23.85 ± 3.19
Line 296: [1,6,7,25,26,38] - you don't need so many citations
Author Response
Reviewer #1:
Thank you so much for your review, kind comments, and valuable suggestions. We have modified the text according to them.
The manuscript entitled Structural changes, electrophoretic pattern, and quality attributes of camel meat treated with fresh ginger extract and papain powder has great scientific potential. Some major changes need to be made.
Query 1: Page 2, line 68:reveal - incorrectly used word.
Answer 1: The word revealing was replaced by examination of (line 68 in the revised manuscript)
Query 2: Page 2, line 81: semitendinosus – italics
Answer 2: Corrected
Query 3: Page 2, line 90: semitendinosus – italics (correct throughout the manuscript)
Answer 3: Corrected throughout the manuscript
Query 4: Page 3, line 111: 2.5. Investigations 2.5.1. Physicochemical Criteria - incorrectly used word
Answer 4: Corrected and replaced by the term quality attributes throughout the manuscript and in Table1 & Table 3.
Query 5: Page 5, line 223: [7,24,25,26,27] - you don't need so many citations
Answer 5: The use of citations has been corrected according to the journal requirement.
Query 6: Table 1: Collagen solubility (%) - what is the reason for such a big difference in collagen? Control 3.60c ± 0.19, 5% ginger+ 0.5%, Papain 23.85 ± 3.19
Answer 6: Thank you for your comments. During tenderization, the proteolytic enzymes in ginger and papain attack all muscle proteins including connective tissue. Additionally, increased collagen solubility of samples treated with ginger (10.36 ± 2.67) and papain (18.28 ± 2.70) might be due to an increase in permeability of the connective tissue, which will disintegrate easily. Furthermore, proteases may also promote structural alterations through action on intermolecular cross-links. Nonetheless, mixtures of both enzymes (ginger and papain) induced the most significant higher collagen solubility (23.85 ± 3.19) due to the combination effect with consequence producing significant difference as compared with the control untreated sample.
Query 7: Line 296: [1,6,7,25,26,38] - you don't need so many citations
Answer 7: The reference numbers 7, 25 and 38 were deleted from this sentence.
Reviewer 2 Report
There are few comments and suggestions for consideration:
1. Materials and Methods
- Line 90: Why the Authors use the term “Camelus dromedaries” and not the Latin name of the species Camelus dromedarius L. (1758)?
- Line 99: Why did you use so small size samples (3 cm3)? If the Authors cut muscle into uniform-sized chunks (3 × 3 × 3 cm), (line 96), sample should be size 27 cm3.
- Lines 171-173: If the Authors used the same heat treatment to prepare the samples for texture analysis and determination of cooking loss, description of heating treatment should be moved to the subsection 2.5.1.6.
2. Results and discussion
- There is one empty page between lines 223 and 224.
- Table 1: check units - TBARS is mg/Kg, should be mg/kg, shear force is Kgf, should be kgf, but kgf does not follow the International System of Units (SI). SI fore unit is N (Newton). Please correct the data (1 kgf = 9,80665 N).
- Table 1: there are no measure unit for bacterial counts.
- Table 1: APC is plate count agar which is bacteriological medium used for determination of the total number of live, aerobic bacteria in a sample, but it is not the total number of live, aerobic bacteria.
- Fig 1: black arrows do not indicate muscle fiber but fiber breaks (damages).
- Lines 318-320: “…, with increased muscle fibers spaces, …”, should be “increased spaces between muscle fibers”.
- Lines 331: lack of full names for abbreviations (MFD and SL).
- the separation of microscopic photos into 2 figures (fig. 3 and 4, fig. 5 and 6) makes it difficult to compare and analyse
Author Response
Reviewer #2:
Thank you so much for the critical revision and the valuable suggestions.
There are few comments and suggestions for consideration:
Query 1: Materials and Methods: Line 90: Why the Authors use the term “Camelus dromedaries” and not the Latin name of the species Camelus dromedaries L. (1758)?
Answer 1: Thank you for your comments. The term “Camelus dromedaries” was corrected to “Camelus dromedaries L (1758)” in the materials and methods section (lines 82 and 92) in the revised manuscript.
Query 2: Materials and Methods: Line 99: Why did you use so small size samples (3 cm3)? If the Authors cut muscle into uniform-sized chunks (3 × 3 × 3 cm), (line 96), sample should be size 27 cm3.
Answer 2: Special thank for this suggestion! The correct version has been replaced in the text (27 cm3)
Query 3: Materials and Methods: Lines 171-173: If the Authors used the same heat treatment to prepare the samples for texture analysis and determination of cooking loss, description of heating treatment should be moved to the subsection 2.5.1.6.
Answer 3: The description of heating treatment was added in subsection 2.5.1.6.
Query 4: Results and discussion: There is one empty page between lines 223 and 224.
Answer 4: Thank you for your comments. This empty page appears only in the pdf document generated by the submission system however, in the word file there is no empty page.
Query 5: Results and discussion: Table 1: check units - TBARS is mg/Kg, should be mg/kg, shear force is Kgf, should be kgf, but kgf does not follow the International System of Units (SI). SI fore unit is N (Newton). Please correct the data (1 kgf = 9,80665 N).
Answer 5: Thank you so much for your review. The unit mg/Kg was corrected to mg/kg in Table 1. Furthermore, the data of shear force value was changed from the unit kgf to the unit of N (Newton) in Table 1.
Query 6: Results and discussion: Table 1: there are no measure unit for bacterial counts.
Answer 6: The measuring unit for bacterial counts (Log10 CFU/g) was added in Table 1.
Query 7: Results and discussion: Table 1: APC is plate count agar which is bacteriological medium used for determination of the total number of live, aerobic bacteria in a sample, but it is not the total number of live, aerobic bacteria.
Answer 7: In order to clarify this issue the APC has been replaced in the Table 1 with Aerobic Plate Count (APC)
Query 8: Results and discussion: Fig 1: black arrows do not indicate muscle fiber but fiber breaks (damages).
Answer 8: The word fiber breaks was added in Fig 1.
Query 9: Results and discussion: Lines 318-320: “…, with increased muscle fibers spaces, …”, should be “increased spaces between muscle fibers”.
Answer 9: Corrected
Query 10: Results and discussion: Lines 331: lack of full names for abbreviations (MFD and SL).
Answer 10: The full names for abbreviations of MFD (muscle fiber diameter) and SL (Sarcomeres length) were mentioned in the materials and methods section lines 125 and 126.
Query 11: Results and discussion: the separation of microscopic photos into 2 figures (fig. 3 and 4, fig. 5 and 6) makes it difficult to compare and analyse
Answer 11: The 4 figures were placed on a single page, and the two subchapters (Transmission electron microscope and scanning electron microscope subchapters were joined together)
Reviewer 3 Report
The authors have submitted an article that outlines an interesting investigation on the structural changes, electrophoretic pattern, and quality attributes of camel meat treated with fresh ginger extract and papain powder. The results of this study are novel and are of great importance to meat industry and consumers.
I would recommend a several minor changes. My comments and suggestions are outlined in the submitted PDF file.

Author Response
Reviewer #3:
Great thanks for your revision and respected suggestions.
The authors have submitted an article that outlines an interesting investigation on the structural changes, electrophoretic pattern, and quality attributes of camel meat treated with fresh ginger extract and papain powder. The results of this study are novel and are of great importance to meat industry and consumers.
Thank you very much for your positive comments and appreciation! We are delighted to read these lines! Thank you so much!
I would recommend a several minor changes. My comments and suggestions are outlined in the submitted PDF file.
In the revised version of the manuscript, we have addressed all of your comments and suggestions that were highlighted during the revision process in the pdf version of the submission.
Thank you again!